# Extracellular Vesicles as Mediators of Cancer Disease and as Nanosystems in Theranostic Applications

**DOI:** 10.3390/cancers13133324

**Published:** 2021-07-02

**Authors:** Renato Burgos-Ravanal, América Campos, Magda C. Díaz-Vesga, María Fernanda González, Daniela León, Lorena Lobos-González, Lisette Leyton, Marcelo J. Kogan, Andrew F. G. Quest

**Affiliations:** 1Laboratorio de Comunicaciones Celulares, Centro de Estudios en Ejercicio, Metabolismo y Cáncer (CEMC), Programa de Biología Celular y Molecular, Facultad de Medicina, Universidad de Chile, Santiago 8380453, Chile; raburgos@uc.cl (R.B.-R.); ame.campos@ug.uchile.cl (A.C.); magdiaz@ug.uchile.cl (M.C.D.-V.); mariagonzalez1@ug.uchile.cl (M.F.G.); lleyton@med.uchile.cl (L.L.); 2Centro Avanzado para Estudios en Enfermedades Crónicas (ACCDIS), Santiago 8380453, Chile; daniela.leon92@ucuenca.ec; 3Exosome Biology Laboratory, Centre for Clinical Diagnostics, UQ Centre for Clinical Research, Royal Brisbane and Women’s Hospital, The University of Queensland, Brisbane 4029, Australia; 4Grupo de Investigación en Ciencias Básicas y Clínicas de la Salud, Pontificia Universidad Javeriana de Cali, Cali 760008, Colombia; 5Departamento de Química Farmacológica y Toxicológica, Facultad de Ciencias Químicas y Farmacéuticas, Universidad de Chile, Santos Dumont 964, Independencia, Santiago 8380494, Chile; 6Centro de Medicina Regenerativa, Facultad de Medicina, Universidad del Desarrollo-Clínica Alemana, Santiago 7590943, Chile; llobos@udd.cl

**Keywords:** extracellular vesicles, hallmarks of cancer, drug resistance, theranostics

## Abstract

**Simple Summary:**

Cancer is the second leading cause of death in humans, and in 2020, 9.8 million cancer-related deaths were reported worldwide. In the last 20 years, it has become apparent that small vesicles released by cancer cells, referred to as extracellular vesicles (EVs), are key players in cell–cell communication in the tumor environment, and as a consequence, research in this area has increased dramatically. This review summarizes the recent advances in our understanding of how EVs serve as mediators of communication between cancer cells and with their surroundings in order to promote the acquisition of specific characteristics that permit their aberrant behavior. In addition, we dwell on how EVs aid in the development of drug resistance, which is a frequent cause of treatment failure in chemotherapy. Finally, we discuss an exciting new area of research that envisions harnessing the unique characteristics of EVs for therapeutic and diagnostic purposes (theranostics). Taken together, the available literature suggests that advances in our understanding of EV biology in the next decades will likely be critical to achieving more effective treatments in cancer patients.

**Abstract:**

Cancer remains a leading cause of death worldwide despite decades of intense efforts to understand the molecular underpinnings of the disease. To date, much of the focus in research has been on the cancer cells themselves and how they acquire specific traits during disease development and progression. However, these cells are known to secrete large numbers of extracellular vesicles (EVs), which are now becoming recognized as key players in cancer. EVs contain a large number of different molecules, including but not limited to proteins, mRNAs, and miRNAs, and they are actively secreted by many different cell types. In the last two decades, a considerable body of evidence has become available indicating that EVs play a very active role in cell communication. Cancer cells are heterogeneous, and recent evidence reveals that cancer cell-derived EV cargos can change the behavior of target cells. For instance, more aggressive cancer cells can transfer their “traits” to less aggressive cancer cells and convert them into more malignant tumor cells or, alternatively, eliminate those cells in a process referred to as “cell competition”. This review discusses how EVs participate in the multistep acquisition of specific traits developed by tumor cells, which are referred to as “the hallmarks of cancer” defined by Hanahan and Weinberg. Moreover, as will be discussed, EVs play an important role in drug resistance, and these more recent advances may explain, at least in part, why pharmacological therapies are often ineffective. Finally, we discuss literature proposing the use of EVs for therapeutic and prognostic purposes in cancer.

## 1. Introduction

Extracellular vesicles (EVs) were initially identified in the 1950s as a type of particle derived from platelets present in plasma [1]. Approximately 20 years later, these particles were still merely considered as “platelet dust” or an insignificant platelet by-product [2]. It took several years before the role of EVs was revealed to be very much the opposite of meaningless cell debris, as their fundamental role in regulating homeostasis at the local and systemic level became apparent [3,4]. EVs are generally described as a heterogeneous population of membrane-enclosed, non-replicating, and sub-micron sized structures, which are actively secreted by a wide variety of eukaryotic and prokaryotic organisms [5,6]. Moreover, EVs can be found in biological fluids, such as serum, plasma, urine, saliva, and breast milk, amongst others [7,8,9,10]. In general terms, EVs can be separated into three subtypes according to their biogenesis and biophysical properties [11], namely exosomes, microvesicles, and other small membrane-limited fragments, such as apoptotic bodies, which are generally thought to be less relevant to cell-to-cell communication [12,13].

Indeed, EVs can also induce important changes in recipient cells [4,14,15]. Specifically in cancer, EVs secreted by tumor cells promote the development of tumor-related features in recipient cells and the acquisition of the cancer hallmarks described in the literature [16]. Furthermore, several studies have documented that cancer cells secrete increased levels of EVs when compared to normal cells [17,18]. Considering the aforementioned data and the fact that EVs play an important role in cancer progression, EVs can also be envisioned as appealing targets for developing non-invasive liquid biopsy strategies in patients with cancer. These micron-sized particles can be readily isolated from biofluids as mentioned, and they can be used to facilitate cancer diagnosis and surveillance. Moreover, they can serve to evaluate treatment efficacy, as well as identify patients prone to cancer relapse and/or resistance to therapy [19,20]. Interestingly, EVs have ultimately been described to display considerable potential as novel transport vehicles, which may be employed to deliver molecules or chemotherapeutic drugs in a targeted manner to tumors. In doing so, toxicity or adverse effects can be reduced in comparison to conventional treatment approaches [21,22]. Thus, this review will discuss literature relating to the role of EVs in promoting acquisition of the hallmarks of cancer and also the use of these vesicles in cancer therapy.

It should be mentioned that one of the many difficulties associated with the EV research field in recent years has been the considerable confusion that exists with respect to their nomenclature. This can be attributed largely to the lack of a consensus between the type of isolation used to purify EVs and the techniques used to distinguish between EV subtypes according to their biogenesis or release. To tackle this problem, several EV researchers decided to combine their knowledge to unify the currently used nomenclature [15,23]. This effort gave rise to the development of guidelines, which permit distinguishing between EVs according to their size, density, molecular cargo, or information regarding the cell of origin. In addition, these guidelines also determined that the terms “exosomes” or “microvesicles” should only be used, for example, when imaging techniques were used to confirm a specific biogenesis pathway [15,23]. Thus, in this review, we will refer to the terms “exosomes”, “microvesicles”, or “apoptotic bodies” only when data regarding their biogenesis is presented and confirmed. Alternatively, when such data are unclear or lacking, the term “EVs” will be used instead. In doing so, this review focuses the discussion predominantly, but not exclusively, on the effects of exosomes.

## 2. Extracellular Vesicles

Extracellular vesicles (EVs) are mainly featured as a heterogeneous population of membrane-enclosed, non-replicating, and sub-micron sized structures, which are actively secreted by a wide variety of eukaryotic and prokaryotic organisms [5,6]. In addition, EVs are mediators of communication between cells in physiological and pathological settings, and they transport a diverse array of biomolecules, including lipids, nucleic acids, carbohydrates and proteins [6,24]. Finally, EVs can be sorted into three different subtypes according to their biogenesis and biophysical properties (Figure 1) [11].

### 2.1. Exosome Biogenesis

Exosomes are currently considered the most studied subtype of nano-sized vesicles smaller than 150 nm, which originate by the inward budding of endosomes or multivesicular bodies (MVBs) toward the luminal space, which results in the formation of intraluminal vesicles (ILVs) that are also known as exosome precursors [11,25]. As a next step, these ILV-containing MVBs can either be redirected to degradation in the lysosome or fuse with the plasma membrane (PM), thus leading to the release of exosomes into the extracellular space [6,11]. Interestingly, exosome biogenesis and cargo sorting are closely related processes. In this regard, there are two well-known mechanisms of ILV formation that may depend or not on the presence of a particular set of Endosomal Sorting Complexes Required for Transport (ESCRT), namely complexes 0, I, II, and III [6,11]. The first mechanism involving ILV formation requires the presence of ESCRT-0, which has been described to select ubiquitinated proteins and segregate them into microdomains found on the endosomal membrane. In addition, ESCRT-I and II are held responsible for the binding of specific cargoes to the aforementioned microdomains. Subsequently, these complexes recruit the Alix protein, which aids in recruiting the ESCRT-III complex containing proteins involved in the last stages of ILV formation or vesicle budding and complex detachment from the endosomal membrane [11]. The second mechanism, also considered as being independent of ESCRT, requires the presence of Alix and transmembrane proteins, such as syntenin and syndecan, which are responsible for recruiting specific molecular cargoes (adhesion molecules, growth factors, integrins, etc.) along with the tetraspanin CD63, which eventually leads to ILV formation [11]. Recent evidence points towards the existence of a third mechanism of ILV biogenesis, which does not depend on components of the ESCRT complexes but rather involves the participation of membrane lipid microdomains or lipid rafts. The specific characteristics of the lipids involved favor inward bending of the MVB membrane and thereby promote ILV formation [11]. One of the main proteins involved in this lipid-dependent mechanism is the neutral sphingomyelinase, which is responsible for generating ceramide, a conical lipid with a small head group that favors bending toward the lumen of the MVB membrane [26].

### 2.2. EV Release from the Cell Surface

MVBs can either fuse with the plasma membrane for release of their content or with lysosomes for their subsequent destruction [6,27]. Several reports are available indicating that the final destination depends on factors such as the interaction with microtubules or the actin cytoskeleton, as well as the engagement of specific members of the Rab GTPase family of proteins [27]. Examples in the latter case include Rab27b, Rab11, and Rab35, which promote MVB motility and fusion with the plasma membrane in HeLa, K562 (bone marrow chronic myelogenous leukaemia cells), and Oli-neu (oligodendroglial) cells, respectively [28,29,30].

The second subtype of vesicles ranging in size 50–500 nm (up to 1000 nm), also known as microvesicles (MVs), ectosomes, oncosomes, or microparticles, are described to be released from the cell surface by blebbing from the plasma membrane and subsequent membrane fission [6]. Interestingly, MVs are formed by phospholipid redistribution, positioning phosphatidylserines to the outer leaflet followed by actin–myosin contraction [31,32]. In addition, MV biogenesis requires the participation of small GTPases, such as ADP-ribosylation factor 6 (ARF6) [33,34] and Ras-related proteins, e.g., Rab-22A [34,35]. Importantly, ESCRT complexes also participate in MV formation [32], increasing the level of complexity in EV subtype studies when evaluating vesicle biogenesis. In addition, MV release has been shown to involve Rho family members, such as RhoA, which promotes MV release via ROCK and ERK activation [27,36]. Moreover, RhoA, together with ARF6 and ARF1, increases myosin contractility, thereby favoring MV fission and the subsequent pinching-off from the plasma membrane [6,27,33,37].

Apoptotic bodies, referred to as the third subtype of EVs in the literature, vary widely in size ranging from 50 to 2000 nm in diameter and are ultimately produced by an essential physiological process, which is known as programmed cell death or apoptosis. One of the main features of apoptotic bodies is that mechanisms for specific sorting of organelles, RNA and DNA fragments can be detected, which are absent in other EV subtypes [32,38].

### 2.3. Exomeres

The discovery of exomeres was made possible by the development of new technologies to isolate and visualize EVs. In this regard, two studies report on the efficient isolation of exomeres by optimizing asymmetric-flow field-flow fractionation and ultracentrifugation protocols [39,40]. Exomeres are approximately 50 nm and smaller in size than EVs. In addition, they were shown to be highly enriched in calreticulin, argonaute proteins, amyloid precursor proteins, proteins associated with coagulation (for instance, factors VIII and X), and enzymes involved in metabolism (e.g., glycolysis), especially glycolysis, and mammalian target of rapamycin complex 1 (mTORC1) metabolic pathways [40,41]. Moreover, several recent reports have shown that exomeres can carry nucleic acids, such as DNA, RNA, and miRNAs along with lipids, such as ceramide, esterified cholesterol triglycerides, and phosphatidylcholine [42]. Interestingly, exomeres are not limited by a lipid bilayer, but instead are enriched in certain types of lipids, which differ from those found in exosomes [41]. Although limited information is available concerning their biogenesis, the absence of a lipid bilayer suggests that exomeres cannot be classified as EVs but rather should be viewed as a new type of extracellular particle (EP). In addition, the absence of ESCRT components in these EPs suggests they are different from EVs derived from the plasma membrane or generated via the endocytic pathway [42]. Despite such differences, a novel role for exomeres has been proposed in cancer, since they were shown to promote tumor organoid growth in recipient cells [40].

Interestingly, a novel role for exomeres in the COVID-19 pandemic was suggested, as full-length angiotensin-converting enzyme 2 (ACE2) was reported to be contained in EVs from colorectal cancer cells. Specifically, these cells were able to shed ectodomain fragments of ACE2 that were enriched in exomeres [43]. Given that soluble human recombinant ACE2 can bind to SARS-CoV-2 [44], the binding of SARS-CoV-2 S protein to ACE2 fragments in EVs and exomeres may play an important role in controlling the infection [43]. A relevant question at this point is whether the ability to shed ACE2 fragments is limited to cancer cells and if so, thinking of treatments for SARS-CoV-2 infection, why this might be the case.

### 2.4. EVs in Cell Communication

EVs have emerged as essential players in cell-to-cell communication, because they represent a complex type of “biological package” capable of transporting a wide variety of molecules from one cell to another.

EVs can elicit cellular responses without the need to be internalized into a cell by two mechanisms referred to as soluble and juxtacrine signaling. Soluble signaling involves the proteolysis of an EV surface ligand and its subsequent binding to a cell membrane receptor, whereas juxtacrine signaling requires the juxtapositioning of ligands and receptors on opposing surfaces of the EVs and the target cell [45].

On the other hand, EV internalization by recipient cells involves at least four mechanisms: membrane fusion, phagocytosis, micropinocytosis, and endocytosis. For membrane fusion, the EV membrane directly merges with the cell plasma membrane and transfers cargo molecules to recipient cells. Protein members of the Rab family, Sec1/Munc-18-related proteins (SM proteins), Lamp-1, and SNAREs contribute to this process. Uptake by phagocytosis inevitably results in the fusion of the phagosome with the lysosomes and the degradation of EV content. Phagocytosis likely represents a process important for EV clearance by the immune system, given that the presence of phosphatidylserine (PS) on the outer EV surface promotes their uptake. Indeed, PS appears to represent an essential component of EVs for triggering their clearance by phagocytosis. Macropinocytosis is characterized by plasma membrane ruffling induced by growth factors or other signals. The resulting vesicles contain extracellular fluid and small particles. Macropinocytosis is induced by signaling cascades involving Rho family GTPases, which facilitate actin-driven membrane protrusion formation. The mechanism of EV macropinocytosis is dependent on Na+/ H+ exchanger function, actin, Rac1 GTPase activity, cholesterol, dynamin, and low pH. Endocytosis is divided into two types of receptor-mediated processes: clathrin-mediated endocytosis (CME) and caveolin-dependent endocytosis (CDE). CME is produced by the interaction between ligands on the EV surface and specific receptors present on the plasma membrane that utilize clathrin and adaptor protein 2 (AP2) complexes for the subsequent formation of clathrin-coated vesicles (intracellular) to internalize EVs. The clathrin coat alters the structure of the plasma membrane to promote invagination and vesicle fission. Once inside the cell, the clathrin coat of the vesicles is removed to permit fusion with the endosome and transfer of cargo molecules. CDE requires the presence of caveolins, which associate with cholesterol-rich microdomains in the plasma membrane and form small flask-shaped invaginations together with cavins. Hence, CDE is sensitive to cholesterol depletion. In addition, dynamin 2 is a common regulator of endocytosis that has been implicated in CME and CDE [46].

In summary, soluble signaling, juxtacrine signaling, and membrane fusion are more likely to culminate in a cellular response, since EV components do not enter the endosomal-lysosomal degradation pathway directly.

EVs can modify the behavior of recipient cells depending on the biological message or cargo that is being transferred from the donor cell or tissue [47]. Specifically in cancer, EVs have been shown to play a critical role in cell-to-cell communication in the tumor microenvironment that permits the acquisition and maintenance of cell traits, which are referred to as the hallmarks of cancer.

### 2.5. Regulation of EV Release in Cancer

The number of EVs circulating in the blood of patients with different diseases is elevated compared to healthy subjects. For example, patients with breast, ovarian, gastric, prostate, liver, colon, and pancreas cancers have higher levels of exosomes in plasma than healthy donors [48]. Moreover, in gastric cancer, elevated levels of EVs were associated with more advanced stages of disease development [49,50]. In addition, patients with hematological malignancies have higher EV levels compared to healthy controls. Interestingly, among the latter patients, those with Hodgkin lymphomas, multiple myeloma, and primary myelofibrosis had a higher proportion of smaller EVs in blood samples [51], suggesting that vesicle size relates to function. Using scanning electron microscopy, normal human ovarian cells were found to release EVs from a few select areas of the plasma membrane, while ovarian serous adenocarcinoma cells release EVs from the entire cell surface [52]. Elevated EV release in cancer cells has been proposed to occur via a Ca2+-Munc13-4-Rab11-dependent pathway. Specifically, the expression of Munc13-4, a Ca2+-dependent Rab-binding protein, is elevated in cancer cells, which combined with the increased Ca2+ levels enhances exosome release from cancer cells [53]. Comparison of the breast cancer cells MCF-7 and MCF-7 LTED (Long-Term Estrogen Deprived, a cell line model for the resistance to aromatase inhibitors) revealed a significant increase in exosome secretion from the MCF-7 LTED cells. This was accompanied by an increase in Rab GTPase expression, which could represent another mechanism that permits increased exosome release from more malignant cells [54]. Finally, EV release from cancer cells can be increased by microenvironmental factors, such as hypoxia, increased glycolysis, an acidic microenvironment, calcium signaling, and irradiation [55].

## 3. EV-Mediated Function in Cancer

Extracellular vesicles have many physiological and pathophysiological functions. In cancer, EVs play an important role in many, if not all, stages of cancer development, including tumorigenesis, epithelial–mesenchymal transition, metastasis, and drug resistance. The available evidence also indicates that EVs play a role in many types of cancer, including gastric cancer, breast cancer, melanoma, and lung cancer, among many others. Moreover, EVs are involved in the acquisition of all the “hallmarks of cancer” (see Figure 2). Initially described by Hanahan and Weinberg (2000) [56] and updated in 2011 [16], these traits refer to several biological characteristics that are acquired by cancer cells during the multistep process leading to tumor development. In the following section, we will summarize evidence available from in vitro and in vivo studies indicating how EVs participate in these events (see Figure 2).

### 3.1. Sustaining Proliferative Signaling

Cancer cells acquire the ability to proliferate continuously and do so by generating their own signals, thus rendering themselves independent of external input. They may achieve this through a variety of strategies that do not necessarily involve EVs and have been reviewed elsewhere. EVs can promote cell proliferation in an autocrine manner in many types of cancer, including glioblastoma, breast adenocarcinoma, colorectal, and triple negative breast cancer [58,61,88,89]. Specifically, in some cancers, such as bladder, gastric, and non-small cell lung cancer, it has been shown that this increase in proliferation is through the activation of signaling pathways involving phosphatidylinositol 3-kinase (PI3K)/protein kinase B (AKT) or AMP-activated protein kinase (AMPK)/extracellular signal-regulated kinases (ERK) [62,90,91,92]. EVs transfer growth factor receptors, such as the epidermal growth factor receptor (EGFR), which promote receptor-dependent cell signaling [69]. In fact, the EGFR is widely present in EVs from various cancer cell lines [63]. Furthermore, a highly oncogenic isoform of the EGFR (EGFRvIII) is transferred through EVs in glioblastoma, which is a very aggressive cancer disease [58]. Intermediate signaling molecules, such as AKT, PI3K, and cyclins are also found within EVs [59,61,62], and they are likely transferred from cancer cells to target cells to activate proliferative signaling pathways. Nucleophosmin (NPM) is another oncoprotein that is highly enriched in EVs of several cancer cells and participates in many pathways involved in proliferation and growth suppression [63]. Furthermore, EVs may participate in the elimination of tumor suppressors, such as let-7, which is a microRNA precursor highly expressed in EVs from cancer cell lines [60].

The tumor microenvironment, which includes cells such as fibroblasts, myofibroblasts, endothelial, pericytes, and immune cells, is also important for tumor growth and development. EVs play an essential role in communication between tumor cells and the tumor microenvironment. For instance, HeLa cancer cell EVs increase Human Umbilical Vein Endothelial Cell (HUVEC) proliferation [93]. In addition, microvesicles from the cerebrospinal fluid of glioblastoma patients enhanced endothelial cell viability in vitro [94]. This is relevant, since angiogenesis promoted by endothelial cell proliferation increases tumorigenicity. Moreover, EVs isolated from non-small cell lung cancer cells (A549) increase the proliferation of the normal fibroblast cell line HLF1 [95]. Cancer cell EVs containing the mRNA for hTERT, the telomerase transcript, induce phenotypic changes, including increased proliferation and extension of the life span in fibroblasts. In addition, EVs isolated from the sera of patients with pancreatic and lung cancer also reportedly contain hTERT mRNA [57]. As previously stated, the inverse scenario has also been observed, namely that EVs from cancer-associated fibroblasts (CAFs) increase the proliferation of pancreatic and oral cancer cells [96,97]. In summary, the evidence presented highlights how EVs from cancer cells may act by several mechanisms in a paracrine manner to change the behavior of neighboring cancer cells or cells of the tumor microenvironment to enhance tumor growth.

### 3.2. Evading Growth Suppressors

Tumor suppressor genes act in many different ways to limit cell growth and proliferation. Thus, because the acquisition of these traits is key to the development and progression of cancer, tumor suppressor function is frequently reduced or eliminated in tumor cells. Some of the best-studied tumor suppressor proteins include the retinoblastoma (RB) protein and p53; both act as central control nodes within two key complementary regulatory circuits that determine whether cells proliferate or, alternatively, induce senescence and apoptosis [16].

Due to their relevance, many mechanisms have been identified that control the expression of these tumor suppressors; yet, to date no reports are available involving either EVs or exosomes in regulation of the RB protein or vice versa, RB in the regulation of EV composition. Alternatively, however, p53 has been shown to regulate the secretion, size, as well as the RNA and protein cargoes of tumor-derived EVs [98]. Proteomics analysis was used to identify proteins secreted in the culture media that are regulated by p53 in response to DNA damage in human non-small lung cancer cells. A more comprehensive analysis showed that exosomes isolated from the culture medium after p53 activation using ionizing radiation (IR) contained transcriptional targets of p53 (Maspin, PGK1, Eno1, and EF-1α), and unexpectedly, proteins encoded by genes that are not transcriptional targets of p53 (Hsp90β and CyPA). A p53-regulated gene product, tumor suppressor activated pathway-6 (TSAP6), was shown to increase exosome production in cells when p53 was activated in response to IR [99]. However, mechanisms that explain how TSAP6 increases exosome secretion have not yet been identified, although p53 is known to control the intracellular vesicle trafficking system by regulating components of the endosomal compartment (see details about EVs biogenesis in Section 1). The activation of p53 directly increases the transcription of the ESCRT-III subunit CHMP4C [100]. The ESCRT-III complex contains oligomers of small α-helical CHMP proteins, of which CHMP4 family members are the most abundant components [101]. ESCRT-III is required for the scission of the intraluminal vesicles (ILVs) into the MVB lumen during exosome biogenesis [6]. Human colorectal cancer cells expressing a dominant-negative mutation of p53 (R248W) were found to secrete exosomes enriched in several microRNAs (miRNAs), including miR-1246. The miR-1246-enriched exosomes are taken up by adjacent macrophages leading to their reprogramming into the anti-inflammatory, tumor-supportive M2-like phenotype characteristic of tumor-associated macrophages (TAMs) [102].

Beyond the ability of p53 to determine EV content, EVs are also known to regulate p53 activity. Bioinformatics analysis of proteome changes in astrocytes treated with glioblastoma (GBM)-derived EVs predicted the inhibition of p53. At the same time, significantly decreased Δ133p53 and increased p53β (truncated p53 isoforms) transcripts in astrocytes exposed to GBM-derived EVs were reported [103]. Changes in both truncated p53 isoforms suggest that astrocytes acquire a Senescence-Associated Secretory Phenotype that modifies the tissue microenvironment by secreting pro-inflammatory molecules, extracellular proteases, and extracellular matrix (ECM) components. In doing so, such “senescent” astrocytes promote tumor progression [103].

Exosomes from colon cancer cells transfected with a shRNA against p53 downregulated p53 expression in fibroblasts and promoted their proliferation. Among the miRNAs in exosomes from p53-deficient colon cancer cells, the upregulation of miR-1249-5p, miR-6737-5p, and miR-6819-5p was observed. Moreover, each of these miRNAs was shown individually to suppress p53 expression in fibroblasts [64]. These results reinforce the notion that p53 plays an active role in the control of exosomal RNA cargos.

### 3.3. Resisting Cell Death

Tumor cells develop strategies that limit or prevent apoptosis to survive and grow. One of these strategies involves EVs, since several studies have shown that EVs play a role in promoting resistance to cell death. Specifically, EVs are known to transport a defined set of miRNAs that transfer the resistance phenotype to sensitive cancer cells by altering cell cycle control and blocking apoptosis [104,105,106,107,108]. One of the anti-apoptotic pathways that has been linked to EV function is the inhibition of the c-Jun N-terminal kinase (JNK) pathway. Bone marrow-derived mesenchymal stem/stromal cell (BMSC)-derived exosomes have been shown to inhibit the JNK pathway and downregulate the expression and phosphorylation of Bcl-2-like protein 11 (Bim) [109]. Moreover, EVs can help prevent apoptosis under cell stress conditions. In EVs obtained from HeLa cervical carcinoma cells exposed to irradiation induced-stress, elevated levels of the inhibitor of apoptosis protein survivin were detected [110]. Finally, it has been reported that EVs derived from both bladder and gastric cancer cells inhibit cancer cell apoptosis by upregulating the expression of Bcl-2 and cyclin-D1 and downregulating Bax and caspase-3 [91,111].

### 3.4. Enabling Replicative Immortality

Cells in most normal cell lineages in the body can only divide a limited number of times, as defined by the “Hayflick” limit [112]. In cells in culture, repeated cycles of cell division induce initially senescence and then the crisis phase, which generally leads to cell death. However, cells that survive this crisis acquire an unlimited replicative potential. This transition is referred to as immortalization and is typical of cell lines that proliferate without developing senescence. The immortalization of cells, as occurs in tumors, is linked to their ability to maintain telomere regions, thereby avoiding senescence or apoptosis, and it is achieved by increasing telomerase expression. The telomeres are multiple tandem hexanucleotide repeats, which shorten progressively in non-immortalized cells after each cell division. Eventually, these regions lose the ability to protect the chromosome ends and generate unstable chromosome patterns that affect cell viability. The length of the telomer regions determines how many successive divisions a cell can undergo before telomeres are eroded and consequently lose their protective functions. Telomerase, the enzyme that adds telomere repeat segments to the ends of telomeric DNA, is almost absent in non-immortalized cells, but it is expressed at significant levels in human cancer cells, where it favors telomere maintenance [16].

As an example, breast epithelial cancer cells were treated with EVs purified from conditioned media of X-ray exposed cells. Compared to control cells, telomerase activity decreased in EV-treated cells. Moreover, exosome treatment with RNase prevented the effect on telomerase activity. These observations suggest that EVs transfer RNA-mediated information relating to telomerase activity between cells; however, unfortunately, this study did not provide any characterization of the exosomes/EVs [113].

The mRNA of the catalytic subunit of the telomerase reverse transcriptase (hTERT) is shuttled in exosomes from cancer cells to fibroblasts that do not express telomerase, where expression of the protein and activity are subsequently detected. Importantly, exosomes from the sera of patients with pancreatic or lung cancer contained hTERT mRNA as well. Telomerase activity induced phenotypic changes in target fibroblasts, including increased proliferation and delayed onset of senescence. In addition, telomerase activity protected the fibroblasts from DNA damage induced by phleomycin [57]. Later studies showed that hTERT is also present in amniotic fluid stem cell-derived EVs [67].

### 3.5. Inducing Angiogenesis

EVs participate in the regulation of pathological angiogenesis, as well as tumor angiogenesis. Hypoxia, a common feature of most solid malignant cancers, is generated by an imbalance between the altered oxygen supply capacity of the abnormal tumor vasculature and increased oxygen consumption of the tumor cells [114]. Therefore, hypoxia is a key driver of tumor angiogenesis [115,116]. Here, it should be noted that exosomes derived from hypoxic colorectal cancer cells promote angiogenesis in vitro and in vivo via Wnt/β-catenin signaling in endothelial cells [117]. In addition, another in vivo study showed that exosomes isolated from hypoxic lung cancer cells contained miR-23a, which increased angiogenesis [118]. In addition, exosomes derived from hypoxic leukemia cells were shown to enhance tube formation by human umbilical vein endothelial cells (HUVECs) via a miR-210-dependent mechanism [119]. Nevertheless, it is important to consider that although hypoxia is important in the development of angiogenesis, it is not the only relevant factor, given that several different pro-angiogenic molecules are present in EVs from tumor cells that are independent of hypoxia.

EVs secreted by cancer cells contain pro-angiogenic mediators, including vascular endothelial growth factor (VEGFA), interleukin-8 (IL-8), interleukin 6 (IL-6) and fibroblast growth factor 2 (FGF2). Moreover, EVs can contain pro-angiogenic miRNAs, such as miR-21, miR-23a, miR-29a, and miR-30 [58,68,69,70,71,72]. Exosomes derived from gastric cancer cells deliver miR-130a to vascular cells to promote angiogenesis and tumor growth by targeting c-MYB both in vitro and in vivo [120]. In addition, exosomes that contain miR-205 from ovarian cancer cells significantly promoted angiogenesis in an in vivo model [121]. Additionally, using an in vivo nude mouse model, pancreatic cancer cell-derived exosomes carrying miR-27a were shown to promote angiogenesis [122]. In addition, recent research showed that the deleted in malignant brain tumors 1 protein (DMBT1) is enriched in EVs compared to the cancer cell of origin [63]. DMBT1 binds to pro-angiogenic factors and promotes adhesion, migration, proliferation, as well as angiogenesis [123]. Cancer cell-derived EVs also contain proangiogenic ECM remodeling enzymes, such as urokinase plasminogen activator (uPA), as well as the MMP2 and MMP9 [68]. Glioblastomas are among the most studied types of tumors known to release EVs carrying potent inducers of angiogenesis in vitro, ex vivo, and in vivo [124] that modify the phenotype of endothelial cells [58,94,125,126]. To date, two mechanisms have been proposed to understand how tumor-derived EVs may promote angiogenesis. First, the uptake by endothelial cells of exosomes derived from cancer cells is known to be increased. In addition, the expression of certain tetraspannins in cancer cell-derived EVs promotes the internalization of EVs by endothelial cells [127,128,129,130,131]. Therefore, EVs stimulate the transcription of genes related to angiogenesis and promote the migration and proliferation of endothelial cells. For example, EVs secreted by cancer cells were reported to transfer mutant EGFR to tumor endothelial cells, promoting mitogenic MAPK and AKT signaling [69]. Second, it has been reported that EVs mediate intercellular communication in the tumor microenvironment through mechanisms other than the transfer of their luminal cargos to recipient cells, in a manner independent on uptake [132,133]. Indeed, a recent study shows that cancer cell-derived EVs stimulate endothelial cell migration via the heparin-bound 189 amino acid isoform of VEGF, which, unlike other common VEGF isoforms, is enriched on the surface of EVs [134].

### 3.6. Invasion and Metastasis

The acquired ability to migrate and invade allows cancer cells to escape from the primary tumor and establish themselves at a new secondary site in a process commonly referred to as metastasis, which is responsible for 70–90% of all cancer-related deaths [16,135]. Numerous in vitro and in vivo studies show that EVs, in particular exosomes, play an important role in cell migration and metastasis in many types of cancer, including breast, glioblastoma, fibrosarcoma, nasopharyngeal, brain, melanoma, and colorectal, among others [89,136,137,138,139]. Some examples of how cancer cell-derived EVs modulate their environment are provided. For instance, the incubation of poorly metastatic B16F1 cells with EVs from the highly metastatic melanoma cell line B16F10 increased B16F1 metastasis to the lung after intravenous injection in mice [136]. Furthermore, exosomes obtained from the sera of prostate cancer patients increased significantly the invasiveness of DU145 prostate cancer cells in vitro compared to cells incubated with exosomes isolated from healthy individuals of the same age [140]. The loss of Rab27a in melanoma cell lines changes the size and protein composition of released exosomes [141]. Rab27a, a protein known to participate in exosome biogenesis [30], is overexpressed in melanomas. In addition, the loss of Rab27a in melanoma cell lines inhibited spontaneous metastasis in vivo, suggesting that Rab27a is important for the pro-invasive effects of exosomes produced by the wild-type cells [141].

EVs can modulate cell migration and metastasis through a variety of different mechanisms. These include the transfer of molecules that enhance migration, EMT-related molecules, MMPs, and miRNAs [14,73,76,78,142]. For instance, exosomes from a colorectal cancer cell line (HT-29), with high potential to induce liver metastasis, significantly increased in vitro migration and metastasis to the mouse liver of human colorectal Caco-2 cancer cells, which is a cell line with very low metastatic potential to the liver. This effect was proposed to be mediated by elevated levels of the C-X-C Motif Chemokine Receptor 4 in the exosomes [78]. In breast cancer, EVs from the highly metastatic cell line MDA-MB-231 containing caveolin-1 enhanced the migration and invasion in vitro of the less metastatic breast cancer cell line T47-D lacking caveolin-1 [14], providing evidence for the importance of caveolin-1 in the genesis of exosomes with elevated malignant potential. In prostate cancer, Integrin subunits α3 and β1, Talin 1, and Vinculin, proteins all relevant to migration and invasion, were more abundant in EVs of the more aggressive PC3 cell line, compared to the exosomes from less aggressive LNPaC cells. Furthermore, EVs derived from each of these cell lines increased the invasion of non-cancerous cells, which was prevented when integrin subunit α3 was blocked. Additionally, integrin subunits α3 and β1 are increased in EVs isolated from the urine of metastatic prostate cancer patients [142]. Interestingly, stromal cells from gastrointestinal tumors release exosomes containing the receptor tyrosine kinase proto-oncogene KIT (also called CD117), which increases MMP1 expression in smooth muscle cells, creating a positive feedback loop between stromal and tumor cells that favors tumor cell invasion [74]. Moreover, fibrosarcoma exosomes containing fibronectin, an important ECM protein, promoted cell adhesion and migration [143]. Finally, prostate cancer cells produce large oncosomes containing bioactive MMPs, in addition to other molecules that are important for cancer progression, such as caveolin-1 and ADP ribosylation factor 6 [144].

On the other hand, RNA-bearing exosomes are also important for cell migration and metastasis. In particular, some miRNAs are relevant in this context, such as miR-9, -145, -21, -29a, -494, and -542-3p. These miRNAs affect the expression of many different targets, such as cell–cell adhesion molecules, chemokine ligands, cell cycle regulators and angiogenesis-promoting proteins, which are all factors that contribute to metastasis [76,79]. Exosomes derived from primary lung tumors, carrying small nuclear RNAs, were shown to activate Toll-like receptor 3 (TLR3) in lung epithelial cells, inducing chemokine secretion and neutrophil recruitment to the lung that favors the formation of pre-metastatic niches in vivo [145]. In addition, exosomes derived from breast cancer cells were shown to transfer miR-105 to HUVEC cells, where miR-105 reduces the expression of the tight junction protein ZO-1 to promote vascular permeability that favors the spread of cancer cells [146]. Furthermore, exosomes from B16-F10 cells can also induce vascular leakiness, as evidenced by increased pulmonary endothelial permeability [147]. This was corroborated by Hoshino et al. using exosomes with tropism to the lung from the MDA-MB-231-derived human breast cancer cell lines 4175 and 1833, in a mouse model [77]. Taken together, this evidence suggests that exosomes initially increase vessel permeability in order to prepare the pre-metastatic niche.

Moreover, exosomes can also act as a scaffold for the attachment of metastatic cells [143]. In this respect, an interesting study shows that exosome release is important for autocrine cell migration. Specifically, using the chick embryo chorioallantoic membrane assay, as well as in vitro assays, the authors found that exosomes from H10T80 human fibrosarcoma cells enhanced directional migration and promoted adhesion assembly in an autocrine manner. Moreover, in these in vitro assays, exosomes promoted the migration of H10T80 cells by enhancing adhesion. Somewhat surprisingly, miR-210-containing exosomes from HCT-8 colon cancer cells with a more adhesive phenotype inhibited the MET and cell-surface adhesion of a subpopulation of HCT-8 cells with elevated metastatic potential in vitro [148]. These results suggest that exosomes may also reduce the adhesion of tumor cells and thereby favor their dissemination.

Exosomes can also function as vectors that sequester molecules to reduce their intracellular bioavailability, thereby altering the phenotype of the parent cell [75]. For example, the Let-7 and miR-200 miRNA levels observed in exosomes from the ovarian cancer cell lines SKOV-3 and OVCAR-3 were elevated compared to the intracellular levels. This is relevant, given that the let-7 miRNA family suppresses cell proliferation, while the miR-200 family suppresses EMT [75]. Thus, the elimination of these miRNAs through exosomes reduces their intracellular levels.

Regarding the role of exosomes in preparing the metastatic niche and colonization, Hood et al. provided evidence for the importance of melanoma-derived exosomes in promoting metastasis to lymph nodes in vivo. To this end, C57BL/6 mice were pre-conditioned by injecting into the left footpad exosomes isolated from B16F10 cell culture supernatants. The subsequent injection of B16F10 cells into the left footpad revealed that preconditioning increased melanoma cell recruitment to lymph nodes of the mice, which is a preferential site for melanoma metastasis [149]. Another study showed that the intravenous injection of B16F10-derived exosomes following orthotopic injection of B16F10 cells into C57BL/6 mice increased metastasis to the lung. Furthermore, the transplantation of bone marrow-derived cells (BMDC) treated with exosomes derived from B16F10 cells, after subcutaneous implantation with B16F10 cells, resulted in higher metastatic burden in vivo in the lung and ipsilateral lymph nodes, which was attributed to transfer of the MET oncoprotein [147]. In addition, during colonization, exosomal integrins are important for specific organ tropism. In particular, using a knock-down strategy, exosomes containing the α6β4 integrin were shown to promote lung metastasis, while αvβ5 integrin presence was linked to liver metastasis. Furthermore, exosomal integrins were associated with the increased expression of genes related to metastasis, such as S100A8 and S100P, as well as elevated levels of the src protein and phosphorylation on Tyr-416 [77].

Finally, another mechanism by which exosomes can promote metastasis is by reducing the permeability of the vascular endothelial barrier, which is a topic that will be addressed further on in an independent section.

### 3.7. Genome Instability and Mutation

In physiological conditions, the genome maintenance systems find and repair defects in the DNA, maintaining very low rates of spontaneous mutation. Cancer cells often increase the rates of mutation through increased sensitivity to mutagenic agents or deregulation of components of the genome maintenance and repair machinery, or both [16]. Since the late 1990s, several types of defects that affect components of the DNA maintenance machinery, have been described [150]. For instance, DNA repair genes and mitotic checkpoint genes, such as the MutL homolog 1 (MLH1), the breast cancer susceptibility gene 1 (BRCA1), MYH (also known as MUTYH), and the xeroderma pigmentosum group A (XPA) all encode proteins that help to maintain genomic stability [151].

The MLH1 protein is one of seven DNA mismatch repair proteins (MLH1, MLH3, MSH2, MSH3, MSH6, PMS1, and PMS2) in humans. A heterodimer between MSH2 and MSH6/MSH3 first recognizes the DNA mismatch. The MSH2–MSH6 heterodimer allows the binding of a second heterodimer of MLH1 and PMS2/PMS3/MLH3. This protein complex formed between the two sets of heterodimers enables the initiation of repair of the mismatch defect in DNA [152]. EVs isolated from sorafenib-resistant renal cell carcinoma (RCC) cells contain high levels of the microRNA miR-31-5p. Treatment with miR-31-5p-containing EVs suffices to downregulate MLH1 expression in target cells [80]. This mechanism would presumably reduce the activity of the DNA mismatch repair system and lead to long-term accumulation of mutations, but this hypothesis has not yet been corroborated.

BRCA1 promotes the repair of DNA double-strand breaks (DSB) by homologous recombination. BRCA1 associates with BRCA1-associated RING domain protein 1 (BARD1) and other tumor suppressor proteins to initiate the nucleolytic resection of DNA lesions and the recruitment and regulation of the recombinase RAD51 [153], which catalyzes the insertion of single-stranded DNA (ssDNA) into sister chromatids. Using sister chromatid as the template, ssDNA is elongated, and junctions are formed between the two sister chromatids [154]. Recent studies show that BRCA1-deficient fibroblasts treated with uveal melanoma-derived and colorectal cancer-derived EVs transfer malignant traits to target cells, and the authors suggest that BRCA1 activity is necessary to prevent the detrimental effects of cancer-derived EVs in non-cancer cells [155,156].

To date, a literature search for evidence linking EVs to the control of the other two caretaker genes, MYH/XPA, did not yield any results.

### 3.8. Tumor-Promoting Inflammation

Cancer cell-derived EVs promote the generation and persistence of the inflammatory environment, which contributes to disease progression. Fabri et al. demonstrated that the miR-21 and miR-29a contained in exosomes derived from lung cancer cells bind to members of the Toll-like receptor (TLR) family on immune cells. TLR engagement triggers the activation of nuclear factor kappa-light chain-enhancer of activated B cells (NF-κβ), secretion of pro-metastatic inflammatory cytokines, and the transcription of genes that favor tumor proliferation and metastasis [81]. Another study showed that when monocytes are stimulated with EVs derived from oral squamous cell carcinoma (OSCC), the uptake of these EVs by monocytes leads to NF-κB activation and the generation of a pro-inflammatory environment, which was characterized by elevated levels of IL-6, monocyte chemoattractant protein 1 (MCP1), prostaglandin E2 (PEG2) and MMP9 [157]. Using RNA sequencing and proteomics analysis, Haderk et al., observed that expression of the Y RNA (small non-coding RNA) hY4 is increased in exosomes isolated from chronic lymphocytic leukemia (CLL) cells and from the culture supernatant of a CLL cell line. Additionally, when monocytes were treated with these exosomes, PD-L1 expression and cytokine release were induced, facilitating cancer-related inflammation [82].

In addition, the pro-inflammatory effects of tumor-derived exosomes that affect macrophage performance have been described. Wu et al., found that exosomes derived from gastric cancer cells induced macrophages to express higher levels of pro-inflammatory factors, such as IL-6 and TNF-α. These exosomes markedly increased the phosphorylation of NF-κB in macrophages and, additionally, activated macrophages in human peripheral blood monocytes via NF-κB [158]. Moreover, lung cancer cell-derived exosomes transform naïve mesenchymal stem cells (MSCs) into a new kind of pro-inflammatory MSCs (P-MSCs) by activating TLR2/NF-κB signaling [159]. Recently, Pritchard et al. reported that lung tumor cells secrete exosomes that are taken up by macrophages and differentiate into tumor-associated M2 macrophages, which can promote inflammation in the tumor environment and immune suppression [160]. Together, these studies highlight how EVs play an important role as messengers in the communication between tumor cells and cells of the immune system. Such cell–cell communication promotes the genesis of a pro-inflammatory environment that permits the escape of tumor cells from destruction by the immune system.

### 3.9. Deregulating Cellular Energetics

Cancer cells exhibit remarkable metabolic plasticity that is necessary to generate energy and at the same time satisfy the biosynthetic requirements, which permit maintaining proliferation and/or metastatic spread [161]. In addition, particularly for cancer cells in a hypoxic environment, the enzymes of the glycolytic pathway are upregulated, and elevated release of lactate and pyruvate is observed, which leads to an acidification of the tumor environment [162]. In turn, the decrease in pH is associated with an increase in the secretion and uptake of EVs [163] that contain proteins involved in metabolism and miRNAs that target proteins related to metabolic activities of the cell [83,164]. Fatty acid synthase (FASN), a key enzyme involved in the de novo synthesis of FAs, is one of the most frequently identified proteins in EVs [83]. Additionally, not only the protein but also the mRNA of FASN has been identified in prostate cancer (PCa) cell-derived EVs [165], which suggests a possible role for these EVs in the lipogenesis of cancer cells. On the other hand, a study compared by proteomics analysis exosomes from non-aggressive hepatocellular carcinoma cells with those released by aggressive cell lines and found that in the latter case, exosomes are enriched in enzymes involved in glycolysis, gluconeogenesis, and the pentose phosphate pathway [84]. Potentially, these exosomes may be more easily absorbed by the recipient cells, which translates into an increased uptake of these metabolic drivers that affect the metabolic profile of the recipient cells, as is the case for hepatocellular carcinoma cells [84]. However, the presence of glycolytic enzymes in EVs does not necessarily correlate with functional transfer, as shown in a proteomics analysis of adipocyte EVs, which suggested that both glucose oxidation and lactic acid release remained essentially unchanged in recipient cells after treatment with these EVs [166]. Therefore, it will be necessary to increase the number of studies both in vitro and in vivo to establish more conclusively whether EVs enriched in glycolytic enzymes are able to reprogram the metabolism of recipient cells and to what extent this capacity depends on the tumor cell origin.

### 3.10. Avoiding Immune Destruction

Exosomes can induce immune responses by regulating signals controlling both the adaptive and innate immune responses [167]. Tumors avoid being recognized by cytotoxic T cells as a strategy to escape destruction by the immune system. To do so, they can directly impair the functioning of antigen-presenting cells (APC) or cytotoxic T cells, or alternatively induce suppressor T cells. In all cases, efficient immune responses against cancer cells are blocked [168]. Several mechanisms have been described by which EVs participate in the evasion of the immune destruction of tumor cells. For instance, tumor-derived EVs induce immunosuppression by promoting the expansion of regulatory T cells (Treg) and depletion of anti-tumor CD8+ effector T cells, which in conjunction permit tumor escape [169]. Interestingly, metastatic melanomas release EVs, mainly in the form of exosomes, which transport programmed death-ligand 1 (PD-L1) on their surface and suppress CD8 T cell function [86,87]. Recently, a study showed that exosomes from Lewis lung carcinoma or 4T1 breast cancer cells impaired dendritic cell (DC) differentiation and promoted apoptosis [170]. Moreover, several studies have shown that exosomes from cancer cells can inhibit natural killer (NK) cell proliferation and cytotoxic functions, mainly through the downregulation of NK group 2 member D (NKG2D), which is a central mediator of NK cytotoxicity [171,172,173,174,175,176,177]. Xia et al., have shed light on a potentially new mechanism by which cancer-derived EVs may inhibit NK cell activity. Their study shows that exosomes isolated from the supernatants of primary cell cultures of tissue samples from patients with clear cell renal cell carcinoma obtained after nephrectomy display TGF-β1 on their surface, which may impair NK function by activating the Small Mothers Against Decapentaplegic (SMAD) pathway in these cells [85]. Despite their relevance, these results were obtained using in vitro approaches and need to be confirmed in in vivo settings. Elucidating the role of EVs in the evasion of cancer cell destruction by the immune system should aid in the development of new therapies that block evasion of the immune response by tumor cells, consequently enhancing anticancer treatment efficacy.

### 3.11. EVs and Thrombosis

Although the pro-thrombotic role of EVs is not considered a hallmark of cancer, presumably because it does not appear to contribute to cancer development, it does play an important role in determining cancer patient survival and for that reason is considered here.

A variety of studies have identified a role for EVs in modulating processes related to coagulation and hemostasis, as well as in pathologies associated with thromboembolic events, such as sepsis, atherosclerosis and cancer [178,179,180]. Thrombosis is one of the most common complications in cancer patients and represents the second leading cause of death in cancer patients in the United States [181,182,183,184,185]. The procoagulant activity of EVs is associated predominantly with the surface exposure of phosphatidylserine (PS), which facilitates the assembly of complexes, including the coagulation factors VIIIa, IXa, and X, as well as the prothrombinase factors Va, Xa, and II on the EV surface [179]. Moreover, tumor cells release EVs with tissue factor (TF) on their surface, which activates the extrinsic branch of the coagulation cascade [186,187]. Several in vitro and in vivo studies have linked the expression of TF on EVs to their pro-coagulant potential [188,189,190,191,192]. Indeed, circulating TF-positive EVs (TF+EVs) have been observed in leukemia [193], multiple myeloma [194], breast, pancreatic [195], ovarian [180] and lung [189] cancer.

### 3.12. EVs and Cell Competition

In any given tumor, several different cancer cell subpopulations coexist and, consequently, tumor subclones compete for available resources in a process denominated cell competition (CC). This process determines the relative fitness in neighboring cells and permits eliminating defective or damaged cells in communities to favor the proliferation and growth of the most competent cells [196]. Given that this will ultimately determine the nature of a tumor, some evidence relating to factors involved in CC mechanisms and the role of exosomes/EVs in that context will be discussed below.

One of the best characterized factors that regulates CC is the transmembrane protein Flower (hFWE). In humans, there are four splice variants of hFWE (1–4), and co-culture studies revealed that cells expressing hFWE2 or hFWE4 proliferate while triggering caspase-dependent apoptosis in cells expressing hFWE1 or hFWE3 [196]. Although this is perhaps one of the clearest examples illustrating how specific molecules participate in CC, there is unfortunately no published information available indicating that hFWE (1–4) are present in exosomes/EVs.

Bone morphogenetic proteins (BMPs) have also been previously associated with CC. In mammals, pluripotent cells with decreased BMP signaling are eliminated in the presence of WT cells [197]. Calcium-dependent activator protein for secretion 1 (CAPS1) protein promotes metastasis in colorectal cancer cells (CRCs), and exosomes derived from CAPS1-overexpressing CRCs increase the migration of normal colonic epithelial cells. Interestingly, proteomics analysis showed that the overexpression of CAPS1 downregulated the BMP4 cargo in exosomes [198]. Thus, these results suggest that CAPS-1 expressing cells restrict BMP4 export using exosomes to upregulate their own signaling or to prevent the functional transfer of this protein to neighboring cells.

Latent membrane protein 1 (LMP1), an oncogenic protein, plays an important role in malignant transformation. In AGS gastric cancer cell populations, LMP1-positive cells decreased gradually with each cell passage when the cells were co-cultured with LMP1-negative cells. The experiments performed to study this phenomenon suggest that LMP1-positive cells stimulate the proliferation of surrounding LMP1-negative, but not LMP1-positive cells, via EV-mediated EGFR activation [199].

YAP is a transcriptional co-activator that does not bind directly to DNA. The phosphorylation of YAP by LATS kinases can either prime the protein for binding to 14-3-3 proteins leading to cytoplasmic sequestration or ubiquitin-mediated protein degradation. Alternatively, however, active (non-phosphorylated) YAP translocates to the nucleus and binds mainly to transcription factors of the TEA domain family (TEAD). In the nucleus, the YAP–TEAD protein complex transcribes genes that control cell proliferation and apoptosis [200]. In co-culture conditions, cells expressing higher levels of YAP have enhanced growth and cause the elimination by apoptosis of cells expressing lower levels of this protein [201]. Wnt5a-enriched exosomes isolated from lymph node metastasis-derived gastric cancer (LNM-GC) cells induced YAP dephosphorylation in bone marrow-derived mesenchymal stem cells (BM-MSCs) [202]. Experiments performed in *Xenopus laevis* embryos have identified human frizzled-5 (hFz5) as the receptor for Wnt5a [203]. Thus, the autocrine stimulation of gastric cancer cells with Wnt5a-containing exosomes could function as an auto-stimulatory mechanism that increases the proliferation of specific subpopulations of cancer cells in metastatic tumors, which are mediated by the activation of hFz5 receptor and YAP-mediated intracellular signaling.

The non-canonical Wnt-planar cell polarity (PCP) pathway does not involve β-catenin but rather controls cell movement through the activation of RHOA, c-Jun N-terminal kinase (JNK), and nemo-like kinase (NLK)-dependent signaling cascades [204]. Exosomes, secreted from human fibroblasts, stimulate breast cancer cell (BCC) protrusive activity, motility, and metastasis via Wnt/PCP signaling in vitro. In orthotopic mouse models of breast cancer, the co-injection of BCCs with fibroblasts dramatically enhances metastasis in a manner dependent on Wnt/PCP signaling in BCCs. Surprisingly, exosome activity in BCCs was shown to be dependent on Wnt11 produced in BCCs. Proteomics analysis revealed that the fibroblast-derived exosomes do not contain Wnt11. The experiments carried out to elucidate the causes of this unexpected observation showed that fibroblast-derived exosomes are internalized by BCCs and then loaded with Wnt11 [205]. These results show how the interactions between different populations of cancer and stroma cells in complex biological systems can lead to modifications in the composition of exosomes/EVs. Finally, the incorporation of Wnt11 into exosomes/EVs may represent a key factor in determining fitness during CC.

Importantly, it should be noted that numerous other proteins found in exosomes or EVs involved in CC mechanisms were already mentioned in the sections on EV-mediated functions in cancer: see EGFR, MAPK (Section 3.5), p53 (Section 3.2), src (Section 3.6), and JNK (Section 3.3).

## 4. EVs in Cancer Drug Resistance

Chemotherapy is widely used to treat cancer, but the effectivity of such therapies is reduced in several types of cancer due to the development of drug resistance, which can be attributed to the activation of intrinsic or acquired mechanisms. Intrinsic resistance refers to the presence of resistance factors in tumor cells prior to chemotherapy that render the treatment ineffective. Acquired resistance, on the other hand, is developed during the treatment of tumors that were initially sensitive and can be caused either by mutations arising during treatment or through adaptive responses [206]. Moreover, tumors are extremely heterogeneous, so drug resistance can arise through the therapy-induced selection of a minor resistant subpopulation of cells that was present in the original tumor [207]. Alternatively, drug resistance can also be acquired by drug-sensitive cells via communication with drug-resistant cells (cancer or stromal) through EV-mediated transfer of resistance factors. Some of these EV-mediated mechanisms of drug resistance will be explored in the next sections.

### 4.1. EV-Mediated Drug Transport

Regardless of the route of anticancer drug administration, these drugs generally need to be taken up by cancer cells because they target an intracellular process. Note that membrane receptor antagonists are exceptions in this respect. The uptake of such drugs by cancer cells may involve active transport mechanisms or rely on simple diffusion because of high membrane permeability. Independent of the mechanism, these drugs must reach a threshold concentration to be effective. However, cancer cells are known to express multidrug resistance (MDR)–ATP binding-cassette (ABC) proteins that export drugs to the extracellular space. These transporters are membrane-bound proteins that consume ATP to eliminate a wide variety of molecules, even against steep concentration gradients [208]. This phenomenon results in decreased intracellular anticancer drug accumulation, which decreases or even abolishes drug effects. In this context, it is important to mention that an alternative drug export mechanism has been described involving EVs to eliminate the drugs in an ABC transporter-independent manner.

Shedden et al., (2003) were the first to report that anticancer drug resistance and the release of EVs could be mechanistically linked. In cancer cell lines, the expression of vesicle shedding-related genes is associated with chemosensitivity profiles. Furthermore, in the breast cancer cell line MCF7, the fluorescent chemotherapeutic agent doxorubicin was incorporated into EVs and released to the culture media [209]. Similarly, in vitro B-cell lymphoma cell lines efficiently extrude doxorubicin in exosomes [210].

Early studies suggested that cisplatin, once inside tumor cells, may be sequestered into acidic vesicles belonging to a secretory pathway. The treatment of human ovarian carcinoma cells with cisplatin showed that the exosomes released from cisplatin-resistant cells contained more than 2-fold higher platinum levels than those released from cisplatin-sensitive cells [211]. Moreover, exosomes released by drug-resistant melanoma cells that were previously treated with a fixed dose of cisplatin in culture contained varying amounts of the drug depending on the pH of the medium, and the level of cisplatin in the exosomes was higher in acidic culture medium [212]. Additionally, it was reported that mouse leukemia cell-derived exosomes can include paclitaxel and, interestingly, that the paclitaxel-containing exosomes reduced the proliferation of a human pancreatic cell line. These observations suggest that exosomes or EVs can be used to package and deliver active drugs [213].

### 4.2. EVs Transport Drug Efflux Pumps

ABC transporters can confer multidrug resistance to tumor cells. In addition, cancer cells can transmit resistance through horizontal transfer using EVs carrying drug efflux pumps. The first evidence for the transfer of ABC transporters between cancer cells was obtained studying human acute lymphoblastic leukemia cells. P-glycoprotein (P-gp)-containing “microparticles” were isolated from drug-resistant cells and then used to treat drug-sensitive cells. The results revealed that P-gp protein transfer coincided with reduced drug accumulation in recipient cells, confirming that the transfer of functional P-gp was mediated by EVs [214]. Later studies showed that exosomes from docetaxel-resistant human prostate cancer cell lines conferred resistance to previously sensitive target cells. In addition, this study revealed that P-gp was only present in exosomes derived from resistant but not docetaxel-sensitive cells [140]. Exosomes derived from doxorubicin-resistant (DXR) osteosarcoma cells are taken up by recipient cells, where they convey a doxorubicin-resistant phenotype. The treatment of doxorubicin-sensitive (DXS) osteosarcoma cells with exosomes derived from DXR cells reduced the sensitivity of the recipient cells to doxorubicin. Moreover, exosomes from DXR cells contain higher mRNA and protein levels of P-gp. In addition, both P-gp mRNA and protein levels increased in cells after treatment with DXR-derived exosomes [215].

P-gp is the best studied drug efflux pump; however, other members of the ABC transporter family have been identified in cancer cell-derived EVs/exosomes, too. GAIP interacting protein C terminus (GIPC) is a protein regulator of autophagy and the exocytotic pathways in cancer. The knockdown of GIPC in pancreatic cancer cells induces the overexpression and incorporation into exosomes of the ATP-binding cassette sub-family G member 2 (ABCG2). This finding opens up the possibility of horizontal transfer of ABCG2 via exosomes mediates drug resistance in pancreatic cancer [216].

In addition, exposure to the chemotherapeutic drug vincristine increases the secretion of ATP-binding cassette sub-family B member 1 (ABCB1)-enriched EVs by inducing dysregulation of the Ras-related proteins Rab8B and Rab5. The transfer of ABCB1 via exosomes helps sensitive cancer cells develop a drug-resistant phenotype [183].

### 4.3. EVs Transfer Pro-Survival Cargos

EV cargoes also include pro-survival factors, which decrease apoptosis sensitivity and increase cell viability, thus leading to resistance to anticancer drugs. Components of the PI3K/AKT pathway, an oncogenic signaling axis involved in cancer cell proliferation and survival, have been reported in EVs. Exosomes derived from HCC cells induced sorafenib resistance in vitro and in vivo by activating the HGF/c-Met/AKT signaling pathway and inhibiting sorafenib-induced apoptosis [217]. Triple negative breast cancer cell lines, resistant to docetaxel and doxorubicin, release EVs that induced resistance to the same drugs in recipient non-tumorigenic breast cells. The treatment with EVs from the resistant cells increased the expression of eight genes associated with the PI3K/AKT pathway [218].

BRAF is a component of the MAPK pathway involved in cell differentiation and survival. BRAF kinase inhibitors, such as vemurafenib and dabrafenib, are used in advanced melanoma treatment. Platelet-derived growth factor receptor β (PDGFRβ) is a receptor tyrosine kinase that induces activation of the PI3K/AKT pathway. Vella et al. showed that PDGFRβ can be transferred to recipient melanoma cells in EVs, resulting in a dose-dependent activation of PI3K/AKT signaling and escape from MAPK pathway inhibition by BRAF [219].

In addition, resistance to apoptosis is an escape mechanism by which cancer cells acquire drug resistance and thus contribute to cancer progression. Cancer-associated fibroblast (CAF)-EVs induced the drug resistance of gastric cancer cells by decreasing cisplatin-induced apoptosis. The proteomics analysis of CAF-derived EVs identified that annexin A6 plays a pivotal role in the drug resistance of gastric cancer cells via the activation of β1 integrin and the downstream intracellular signaling pathways, involving focal adhesion kinase (FAK) and the yes-associated protein (YAP). Consistently, the inhibition of FAK or YAP efficiently attenuated gastric cancer drug resistance in vitro and in vivo [220].

Survivin is a pro-survival protein member of the inhibitor of apoptosis (IAP) family that is present in EVs derived from different tumor types [221]. Paclitaxel treatment of triple negative breast cancer cells induces the secretion of EVs enriched in survivin, which increased the survival of serum-starved, as well as paclitaxel-treated fibroblasts and breast cancer cells [222].

### 4.4. EVs Mediate Drug Resistance via the Transfer of microRNAs

MicroRNAs (miRs) are well-established components of EVs, and their horizontal transfer favors the development of drug resistance. Sorafenib is a kinase inhibitor drug approved for the treatment of primary kidney cancer, advanced primary liver cancer, and advanced thyroid carcinoma. EVs derived from sorafenib-resistant (SR) cells were taken up by sorafenib-sensitive (SS) RCC cells and promoted drug resistance. Elevated miR-31-5p in EVs derived from SR cells downregulated the expression of MLH1, which is a gene commonly associated with hereditary nonpolyposis colorectal cancer in SS cells and thus promoted sorafenib resistance in vitro. In addition, low expression of MLH1 was observed in SR RCC cells and upregulation of MLH1 expression restored the sensitivity of resistant cell lines to sorafenib. Experiments in mice also confirmed that miR-31-5p could regulate drug sensitivity in vivo. Finally, miR-31-5p levels in circulating EVs from the plasma of RCC patients with progressive disease during sorafenib therapy were higher when compared with the levels observed prior to therapy [80].

Exosomes isolated from gemcitabine (GEM)-resistant human pancreatic cancer stem cells (R-CSCs) inhibited GEM-induced cell cycle arrest and apoptosis as well as promoted tube formation and cell migration in drug-sensitive human pancreatic cancer stem cells (S-CSCs). Elevated miR-210 levels were detected in R-CSC exosomes compared to S-CSCs exosomes, and MiR-210 levels in exosomes were dependent on the GEM doses used to treat cells. Moreover, treatment with R-CSC-derived exosomes increased miR-210 levels in recipient cells [223].

The aforementioned studies are only a few recently published examples of the increasing evidence linking cancer drug resistance to the presence of specific miRNA cargos in EVs. A more comprehensive summary of related information can be found in a recent article by Maacha et al. [221].

### 4.5. EV Interference in Immunotherapies

Specific EV surface antigens can be targeted by immunotherapy where they act as a “hunter” in monoclonal antibody-based therapies by diminishing antibody bioavailability. For instance, rituximab (anti-CD20 antibody) binds to CD20 on the surface of EVs and protects targeted lymphoma cells from rituximab-induced toxicity [224]. EVs secreted either by HER2-overexpressing breast carcinoma cells or present in the serum of breast cancer patients bind to trastuzumab. In vitro studies showed that HER2-containing EVs, but not EVs lacking HER2, prevent the reduction in cell proliferation induced by trastuzumab treatment, although no change in HER2 activation status was detected in EV-treated cells by Western blotting [225].

EVs are involved in additional ways in downregulating the immune response. Melanoma patients display different responses to the immune checkpoint inhibitor pembrolizumab (anti-PD-1). The detection of immune checkpoint ligand (PD-L1) on EVs early after therapy is indicative of whether the patients will respond or not to anti-PD-1 therapy. PD-L1 binds to PD-1 receptors on the surfaces of effector T cells, preventing their ability to target tumor cells for destruction. PD-L1 containing exosomes derived from melanoma cells inhibit the proliferation, cytokine production, and cytotoxicity of T cells. Pre-treatment of the exosomes with the anti-PD-L1 antibodies nearly abolished these effects. In vivo studies suggest that exosomal PD-L1 suppresses anti-tumor immunity systemically [86]. In addition, EVs from glioblastoma stem cells were found to contain PD-L1 and inhibit T cell proliferation and antigen-specific T cell responses [226]. These results suggest that by capturing the anti-PD-1 antibodies on their surface, EVs prevent this antibody from accessing the tumor, thereby permitting PD-L1 to bind to PD-1 on T cells and attenuate anti-tumor immune responses.

These findings further extend our understanding of the implications of EVs in the development of the disease. The composition of cancer-derived EVs can regulate patient responses to chemotherapy using one or more of the aforementioned mechanisms. With this in mind, one may predict that EVs will serve to predict or evaluate therapy efficacy, and as such will likely become powerful tools to improve cancer treatment. However, the clinical application of new techniques for rapid EV detection and characterization remains a pending issue.

## 5. EVs in Organ Tropism, Drug Delivery, Imaging and Theranosis

The intrinsic organ tropism of EVs and their potential physiological benefits, combined with drug loading and targeting strategies, provide multiple therapeutic benefits for drug delivery, such as greater cellular uptake and focalization, prolonged circulation time, immunomodulation, biocompatibility, and stability. Furthermore, EVs can be used as biological nanocarriers with the inclusion of active principles, nanoparticles, or imaging agents. As such, they can significantly improve the therapeutic efficacy and selectivity, as well as facilitate the early detection of multiple diseases, including cancer [227]. However, to consider the use of EVs in potential clinical applications, the effects discussed previously relating to the role of EVs in cancer and other pathologies need to be kept in mind.

### 5.1. EV Organ Tropism

EVs have emerged in recent years as potential tools for the delivery of different bioactive agents to target tissues and specific organs [228,229]. In this context, the cellular origin of EVs is key to determining the tropism toward specific organs. For instance, EVs from melanoma cells predominantly accumulate in the lungs, while EVs from dendritic cells tend to accumulate in the spleen [230]. Interestingly, EVs derived from tumor cells reportedly also show selective tropism toward the tumor tissue from which they originated. EVs from brain endothelial cells can cross the blood–brain barrier and accumulate in the brain and brain tumor tissue, while EVs from melanoma cells preferentially target metastatic melanoma tumors [229,231]. However, it is not clear whether the tumor cells from which EVs originate determine alone their tissue tropism. Garofalo et al. [232] observed the in vitro and in vivo targeting and accumulation of lung cancer cell-derived EVs in colon carcinoma cells and vice versa. This may be taken to suggest the existence of a generalized tropism for tumor-derived EVs toward any neoplastic tissue, regardless of the tumor type. Although the molecular basis for EV tropism is not fully understood, there have been some significant advances in discovering molecules involved in this process [233,234]. For example, integrins are cell surface adhesion molecules with a substantial role in determining EV organ tropism, particularly toward the lung and liver. In particular, the expression of α6β4 and α6β1 is important in the EV tropism toward the lungs, while αvβ5 promotes EV accumulation in the liver [77,235]. Exosomes from rat pancreatic carcinoma cells expressing the Tspan8-α4 complex preferentially accumulate in the pancreas and lungs of rats [236]. There is also evidence showing that the cell migration-inducing and hyaluronan-binding protein (CEMIP), which is enriched in exosomes of brain-tropic metastasis-derived MDA-MB-231 breast cancer cells, promotes exosome accumulation in the brain by generating a pro-metastatic environment [237]. Additionally, expression of the programmed death-ligand 1 (PD-L1) in tumor-derived EVs is important for the suppression of T-cell activation and thereby avoiding the immunological anti-tumor responses [87]. These findings further extend our understanding of EV tropism, which opens up novel possibilities for the selective targeting of diagnostic/therapeutic agents to tumors.

### 5.2. EVs as Drug Delivery Vehicles

EVs have become novel biological delivery vehicles for several cargoes, due to the variety of natural properties that they possess. These vesicles have the intrinsic capacity to cross biological barriers and to transport various cargoes, protecting their content from degradation until reaching the target. Depending on their cellular origin, EVs are highly heterogeneous in content, and such variations contribute significantly to their uptake, organ tropism and immunomodulation [238]. EV tropism is determined by the presence on their surface of different adhesion and immunoregulatory molecules, as well as specific cell receptors, which contribute to enhancing their accumulation in specific tissues [239,240]. This characteristic combined with their small size favors EV accumulation in highly vascularized tissues with deficient lymphatic drainage, such as tumors. This phenomenon, referred to as the enhanced permeability and retention (EPR) effect, can be used as a strategy to increase targeting toward tumors [230]. EVs have been widely studied as drug delivery nanocarriers in cancer research, so recent and representative studies for each application of these vesicles in the delivery of proteins, genetic material, and chemotherapeutics drugs will be described (see Figure 3). In this field, Kim et al. developed a formulation of paclitaxel-loaded exosomes by the sonication and conjugation of an aminoethilanisamide–polyethylene glycol (AA-PEG) vector moiety to target the sigma receptor, which is overexpressed by lung cancer cells. The nanosystem (AA-PEG-exoPTX) possesses a remarkable ability to accumulate in cancer cells and demonstrates high anticancer efficacy in a mouse model of pulmonary metastasis [241].

With respect to protein delivery, Aspe et al. [242] engineered EVs from melanoma cells to overexpress survivin-T34A, which is a dominant-negative mutant variant of the inhibitor of apoptosis protein survivin that blocks the protein’s function. Survivin overexpression plays an important role in the development of resistance to both chemo- and radiotherapy in pancreatic cancer. The authors observed that EVs containing either survivin-T34A alone or in combination with gemcitabine increased apoptosis in multiple pancreatic cancer cell lines, as well as enhanced the sensitivity of these cells to gemcitabine.

Beyond such applications, the use of EVs in site-specific drug delivery can be improved by protein engineering and modifying the vesicle surface by attaching additional ligands to improve EV targeting properties and their interaction with tumor cells [243]. For instance, glycosylphosphatidylinositol (GPI) anchored EV proteins such as decay-accelerating factor (known as CD55) were used by Kooijmans et al. [244] to attach anti-epidermal growth factor receptor (EGFR) nanobodies to EVs and thereby improve targeting to EGFR overexpressing epidermoid carcinoma A431 cells. They showed that the GPI-linked nanobodies were successfully displayed on EV surfaces and greatly improved EV binding to tumor cells in a manner dependent on EGFR density.

On the other hand, EVs readily transfer nucleic acids, such as DNA or RNA, to cells where they can cause specific genetic changes. Regarding genetic drug delivery, Kamerkar et al. [245] engineered EVs known as iExosomes derived from fibroblast-like mesenchymal cells loaded by electroporation with siRNA or shRNA specific for the oncogenic GTPase KrasG12D, which is a common mutation in pancreatic cancer. The iExosomes showed enhanced targeting to oncogenic Kras-expressing cells, which was dependent on CD47 and the uptake facilitated by micropinocytosis. Subsequently, the treatment with iExosomes was shown to inhibit tumor growth and significantly increase the overall survival in multiple mouse models of pancreatic cancer.

### 5.3. EV Imaging for Cancer Diagnosis

Regarding the imaging of tumors, one of the major problems is the tremendous spatial heterogeneity combined with temporal variation, which leads to errors in the diagnosis and surgical treatment of tumors and thus represents major causes of therapy failure [246]. Since EVs permit detecting as little as a few hundred cancer cells, their application in cancer imaging represents a promising new approach. By attaching an optical reporter in the nanoscale dimension to the EVs and combining with optical imaging, robust diagnostic and prognostic modalities can be developed [243]. Using such approaches, tumor-targeted EVs can be monitored in real time to check their distribution and identify the precise location of tumors. Fluorescence is generally used for exosome tracking and imaging because of its great versatility and simple application by incubation of EVs with a variety of lipophilic fluorescent markers. In this field, generally small lipophilic fluorescent dyes, such as DiR, DiD and PKH67, have been used to label the membranes of EVs. Although these dyes are useful for distribution studies, clinical applications for diagnosis have yet to be developed [247,248]. Additionally, EV membranes have been labeled with fluorescent proteins, such as green fluorescent protein (GFP) or tandem dimer tomato (td Tomato) [249]. This type of labeling is considered more stable and suitable for evaluation in clinical applications. EVs can also be labeled using luciferase reporters in the cells of origin to produce bioluminescent proteins that are then included in the EVs and permit stable real-time monitoring [250,251].

Another alternative is the use of semiconductor quantum dots as optical reporters. They are more stable and have tunable optical properties that can be used for a wide range of applications, including in vivo imaging and diagnosis. For instance, Zong et al. [252] and Jiang et al. [253] obtained high-resolution images of breast tumor cells or their metastatic activity by loading either silicon or gold-carbon quantum dots, respectively, onto the outer membrane of the exosomes secreted by SKBR3 cells.

Superparamagnetic iron oxide nanoparticles (SPIONs) represent another interesting system for imaging. They have been effectively incorporated into EVs and then tracked in vivo by magnetic particle imaging and MRI, as has been shown for breast cancer [254] and melanomas [255].

Another type of nanomaterial that can be used for EV imaging is gold nanoparticles (AuNPs), which are highly versatile due to their tunability, biocompatibility, and unique optical properties [256]. The AuNP optical properties are due to the interaction of light with the electrons on the surface of the nanoparticles, which produces the collective oscillation of electrons, a phenomenon called surface plasmon resonance (SPR). This phenomenon leads to higher light absorption and scattering efficiency, thus making AuNPs excellent photoacoustic and Raman imaging agents [257,258]. On the other hand, gold exhibits a high absorption coefficient of X-rays, which make AuNPs useful as contrast agents for computerized tomography. AuNPs can be efficiently incorporated into EVs and then used for imaging, as well as tumor ablation in cancer therapy. In this field, Lara et al. [229] developed a double-labeling method to incorporate AuNPs indirectly into EVs by incubating them with cells and isolating them in EVs, which were then labeled with fluorescent dyes. This combination permitted analyzing the vesicle biodistribution and detecting the presence of small metastatic foci in the animal lungs by neutron activation analysis, NIR fluorescence, CT imaging and gold-enhanced microscopy imaging.

The use of EVs in imaging applications has been made possible by exploiting some of their natural properties. In particular, EVs have a mean size of 50–200 nm and can evade clearance by the mononuclear phagocytes, as well as favor passive extravasation in inflamed tissues [259]. The presence of immunomodulatory molecules, such as CD47 and PD-L1 ligand, on the EV surface aids significantly in avoiding phagocytosis and suppressing T cell activation, respectively [87,245]. EVs also possess a “tunable” surface that can be modified by adding targeting molecules, such as antibodies, aptamers, and ligands, all of which can favor specific EV accumulation in tumors, thereby avoiding undesirable off-target effects [260,261]. For these reasons, EVs are nowadays considered very appealing nanoscale tools for use as diagnostic sensors, as well as therapeutic vehicles in oncology.

### 5.4. EVs for Theranostic Applications

With the advances in nanotechnology and thanks to their unique properties, nanoparticles have become a promising tool in many areas in recent years, including theranosis. This novel concept, which combines the use of an agent for diagnosis and therapy in a single formulation, represents a great advance in personalized medicine. Existing evidence points towards the great potential of EVs both as diagnostic biomarkers and therapeutic tools. Such tumor-derived EVs have characteristic proteomic and genomic signatures, indicating that they represent suitable vectors for cancer diagnosis and prognosis [18,262]. In addition, because EVs can transfer various therapeutic compounds, as well as imaging agents, some researchers have proposed to exploit these vesicles as a tool for simultaneous therapy and active diagnosis (see Figure 4). EVs have been proposed as an ideal solution to overcome limitations of inorganic particles, including toxicity, off-target effects and immunogenicity [263].

In this regard, Jia et al. [264] obtained glioma-targeting EVs with diagnostic and therapeutic potential by conjugating RGE, which is a peptide that binds to neuropilin-1 overexpressed on glioma cells, with the EV membrane by applying click chemistry. In addition, superparamagnetic iron nanoparticles (SPION) and curcumin were synchronously loaded into EVs by electroporation. The engineered system efficiently crossed the blood–brain barrier and provided good results for MRI-targeted imaging when applied to glioma cells and in orthotopic xenograft models. Additionally, SPION-mediated magnetic flow hyperthermia and curcumin-mediated effects combined lead to synergistic antitumor activity. Likewise, Wang et al. [234] designed a new platform for tumor-targeted chemo-photothermal therapy and imaging that was based on combining gold nanorods (AuNRs) with exosomes through a donor cell-assisted membrane modification strategy. First, the membrane of the donor cells was modified with RGD peptides and sulfhydryl groups. Then, the isolated exosomes (RGD-Exos-SH) were functionalized with AuNRs by the formation of Au-S bonds and coupling folic acid (FA) to improve the uptake efficiency by tumor cells. Further, doxorubicin (DOX) was loaded into exosomes by electroporation. Such designer exosomes showed effective accumulation at target tumor sites via dual ligand-mediated endocytosis, which were monitored in nude mice bearing tumor cell xenografts, using non-invasive near-infrared optical imaging. Moreover, the localized hyperthermia induced by the conjugated AuNRs during near-infrared irradiation increases the permeability of exosome membranes to enhance drug release, thereby preventing tumor relapse in a programable manner. Hence, the compatibility of EVs with different therapeutic agents and nanomaterials provides a unique opportunity to develop novel approaches in diagnosis and personalized treatment modalities.

## 6. EVs and Cancer Patient Survival

Cancer-derived EVs/exosomes are promising markers for diagnosis and prognosis in cancer and have been shown to predict the survival of patients. For example, in colorectal cancer, high levels of exosomes in the plasma of patients correlated with elevated presence of the carcino-embryonic antigen (CEA), and such patients tended to have shorter overall survival periods than patients with low exosome levels [265]. In addition, in lung cancer, the presence of the EGFR protein in exosomes from patient plasma has been suggested to represent a biomarker for lung cancer diagnosis [266]. Furthermore, high urinary exosomal levels of the long non-coding RNAs (lncRNAs) MALAT-1 and PCAT-1 correlated with decreased recurrence-free survival in non-invasive muscle bladder cancer (NIMBC) patients [267]. In pancreatic ductal carcinoma (PDAC), the lncRNA Sox2ot was identified in exosomes from plasma samples, and its presence there was closely associated with higher Classification of Malignant Tumors (TNM) stage and reduced overall survival rates of PDAC patients [268]. Combined analysis of exosomal miR-1290 and miR-375 reportedly predicts the overall survival of castration-resistant prostate cancer patients. Over the same follow-up period of 20 months, patients with high levels of both miRNAs had a general mortality rate of 80%, while patients with normal concentrations for both only had a mortality rate of 10% [269]. These are just a few examples from a rapidly growing research field illustrating how exosomes/EVs affect the survival of cancer patients.

On the other hand, tumor-derived exosomes may also be used to evaluate the response to surgery. For instance, their persistence after PDAC tumor resection is related to the presence of hidden metastases. Patients with more than 20% heparan sulfate proteoglycan glypican-1 (GPC1) positive exosomes in peripheral blood have been reported to have lower progression-free and overall survival [270]. Similarly, the detection of high levels of exosome-encapsulated miR-415a was also associated with reduced progression-free and overall survival [271]. In addition, the detection of exosomes containing miR-4525, miR-415a, and miR-21 in the portal vein identified more effectively patients at high risk for recurrence after surgery than did the detection in peripheral blood [272].

In summary, these examples illustrate how the targeted identification of specific proteins or miRNAs in exosomes may serve both diagnostic, as well as prognostic purposes. Indeed, it is important to mention that currently, there are 89 registered clinical trials [273] underway, studying exosomes in cancer patients and looking for markers that could be useful for diagnosis or prognosis. Of these trials, many focus on the prevalent cancer types in the lungs and prostate (14), breast (9), pancreas (8), and colon (6). As the results of these studies become available in the next few years, we may anticipate that a clearer picture should emerge connecting the presence of exosomes and their content to the survival of cancer patients.

## 7. Concluding Remarks

In summary, EVs are a heterogeneous population of membrane-enclosed, non-replicating, and sub-micron sized structures. EVs are actively released by virtually all cell types and by a wide variety of eukaryotic and prokaryotic organisms. EVs can be sorted into three different subtypes according to their biogenesis and biophysical properties: exosomes, microvesicles, and apoptotic bodies (Figure 1). A large number of studies show that EVs are active participants in cell communication. In the context of cancer biology, cancer cell-derived EV cargoes can change the behavior of target cells. The evidence provided shows that EVs are involved in the acquisition of all the “hallmarks of cancer”, that is, biological characteristics acquired by cancer cells during tumor development. Consequently, more aggressive cancer cells can transfer their “traits” to less aggressive cancer cells and convert them into more malignant tumor cells (Figure 2). In addition, EVs play a role in the mechanisms of drug resistance, which can be acquired by drug-sensitive cells through EV-mediated transfer of resistance factors and other mechanisms, which aid in understanding why chemotherapy often fails. However, on the upside, these very characteristics of EVs combined with drug loading and targeting strategies provide unique opportunities for the delivery of different cargoes (Figure 3). Finally, EVs can be used as biological nanocarriers for both therapeutic and diagnostic purposes (theranosis) by including active principles, nanoparticles, as well as imaging agents (Figure 4). With this in mind, it would appear that such “designer” EVs will have a bright future in cancer medicine.

## Figures and Tables

**Figure 1 cancers-13-03324-f001:**
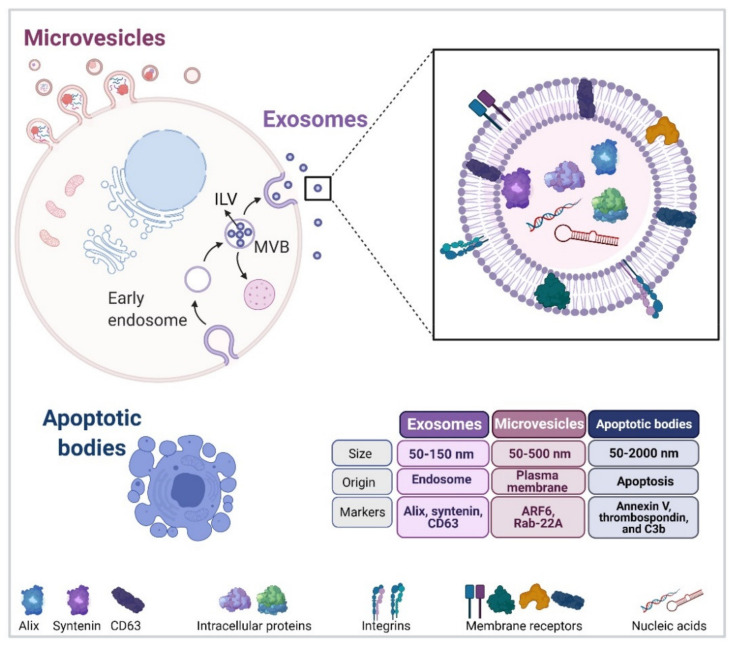
Extracellular vesicles are a heterogeneous population of cell-derived membrane vesicles. Extracellular vesicles (EVs) have classically been divided into three types according to their biogenesis and biophysical properties: exosomes, microvesicles, and apoptotic bodies. Recently, a new group of non-membranous nanoparticles of less than 50 nm, called exomeres, was identified. However, still, little is known about their biogenesis, and proteins that have been connected to exomeres must be characterized further in order to validate them as markers. For this reason, they are not included here. EVs are carriers of a variety of molecules, including proteins, nucleic acids, and lipids. The insert with a close-up view of exosomes shows some molecules commonly transported by them.

**Figure 2 cancers-13-03324-f002:**
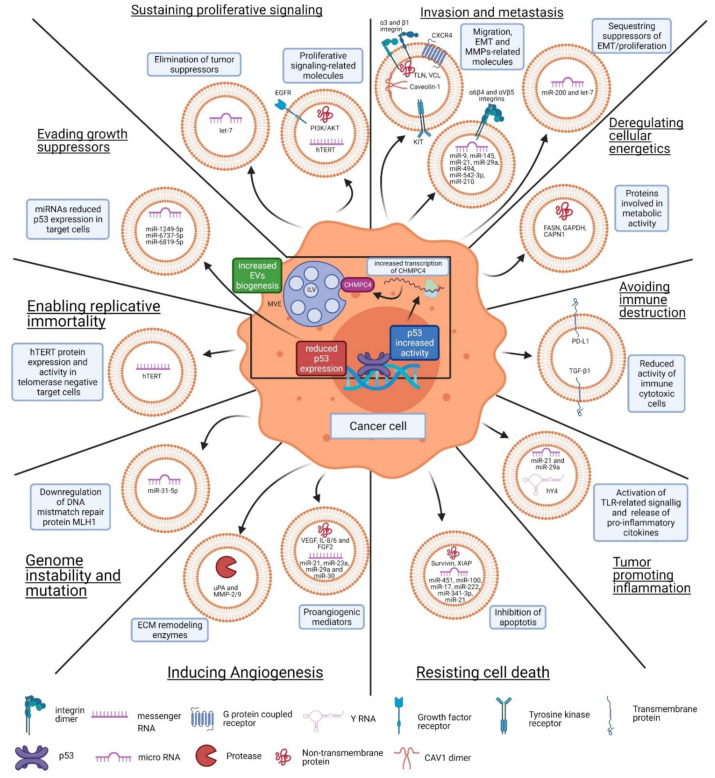
EV-mediated function in cancer. The term “hallmarks of cancer” described by Hanahan and Weinberg [56] refers to ten biological characteristics that are acquired by cancer cells during the multistep process leading to tumor development. The EV-mediated roles reported to date are shown here for each “hallmark of cancer”: Sustaining proliferative signaling [57,58,59,60,61,62,63], Evading growth suppressors [64], Resisting cell death [65,66], Enabling replicative immortality [57,67], Inducing angiogenesis [58,68,69,70,71,72], Invasion and metastasis [14,73,74,75,76,77,78,79], Genome instability and mutation [80], Tumor promoting inflammation [81,82], Deregulating cellular energetics [83,84] and Avoiding immune destruction [85,86,87].

**Figure 3 cancers-13-03324-f003:**
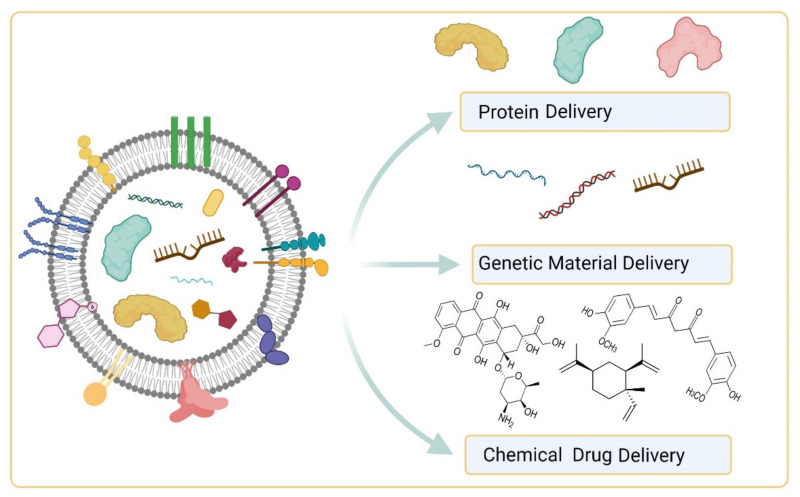
EVs are nanoscale structures with excellent biocompatibility and the ability to transport/deliver many different types of proteins, genetic material, and chemical drugs that can be used in cancer therapy.

**Figure 4 cancers-13-03324-f004:**
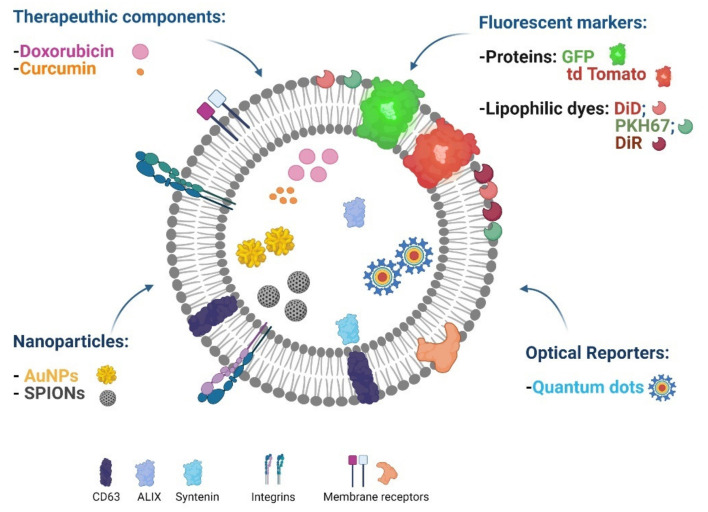
EVs as theranostic nanoplatforms. By combining the natural properties of exosomes with the use of drugs, imaging agents, or NPs, unique platforms can be generated. In combination with different targeting strategies, multiple therapeutic benefits can be achieved, such as improved targeting/uptake, immunomodulation, prolonged circulation time, easy tracking and better therapeutic effects. In doing so, side effects can be reduced, and the need to apply multiple drugs can be eliminated.

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
