# Peer review of "Extracellular Vesicles as Mediators of Cancer Disease and as Nanosystems in Theranostic Applications"

_cancers, 2021, doi:10.3390/cancers13133324_

Round 1
Reviewer 1 Report
This is a truly outstanding review on EVs and their implications in cancer. The authors did an excellent job in discussing how EVs contribute to each of the "hallmarks of cancer" as defined by Hannah and Weinberg, which really makes this review a pleasure to read.
In addition, the authors did a very nice summary on the potential of EVs in drug delivery, imaging, diagnosis and personalized medicine, in particular describing the potential of EVs to transfer therapeutic and imaging agents. I praise the authors for such a comprehensive and timely review.
Reviewer 2 Report
Burgos-Ravanal et al. summarized recent development of cancer EV. This review is comprehensive which included most important studies and update on the field. But I strongly suggested the authors to include their own professional perspective on various parts especially before conclusion, rather than just citing references. Below are other comments.
Major comments:
- Line 31-32. The authors stated that this review is focusing on “how EVs contribute to the acquisition of specific characteristics by cancer cells that permit their aberrant behavior”, which gave an impression that it is addressing the passive effects of cancer cells receiving EV from other cell types.
- Section 2.2. The content is not fully related to the mechanism on how the EV leave cell surface.
- Figure 2. Recommend showing the references in each function they are referring (not just in the text but in the figure as well).
- Section 3. The authors omitted the importance of subclonal competition within tumor. Tumor subclones may compete or cooperate instead of only helping each other. Fitness selection between cells due to tumor evolution may also affect the sorting of specific biomolecules in EV. The authors may consider incorporating this perspective (doi 10.1016/j.tcb.2020.10.002).
- Section 5. The authors may consider adding a subsection reviewing organ specific EV; including the summary of EV-surface markers from different organs.
Minor comments:
- Line 36, the words “next years” maybe too specific?
- Some typos and grammatical mistake scattered in the manuscript, i.e. line 77 bi-ofluid, line 105 can function mediators, line 225, receptor cells…, please proof read.
Reviewer 3 Report
Biology and Pharmacological Effects of Extracellular Vesicles in Cancer by
Renato Burgos-Ravanal et al
This is a review of the literature on EVs in cancer. It is a bibliographic work with 200 references which includes the new results and recent reports. The title of the article is not precise; this is what working model in cancer?
The studies of EVs are quite broad and in the literature we can identify studies of EVs in cellular conditioned medium in vitro and also in plasma in circulating blood.
In this work the authors should specify in the introduction what type of EVs will be described in this review, and discuss on this topic.
One part missing from this review is the the implication of EVs in thrombosis . More than 15% of cancers cause thrombosis and often will be the cause of death in cancer patients. And microvesicles have a central place in the genesis of thrombosis thanks to molecules with pro-coagulant activity such as tissular factor and others.
In this review, it will be interesting to see the manner of the release of EVs, and also up and down regulation of the release of EVs by cancer cells.
This review is interesting but lacks some notion and information specific to the pathophysiology of EVs in cancer and their impact on the survival of cancer patients.
Considering all of these remarks, this review as of today is incomplete and requires major revisions.
Reviewer 4 Report
Simple summary – the authors may want to be more precise in some of the phrasing. “Soon” and “More recently” need to be substituted with more precise timeline. I doubt that small vesicles have been discovered as “key player in cell-cell communication” after 2021, as the current phrasing seems to indicate. Check English and punctuation for correctness.
Abstract
Line 42-44 – Recently discovered exomeres do not have a lipid biolayer. Please incorporate this recent evidence in this paragraph, or remove the reference to lipid bilayer from abstract.
Line 44 – “in recent years” is an understatement, the role of EVs in cell communication has been known for decades now. In general, similar expression referring to timing needs to be more precise throughout the manuscript.
Line 49 – the “Hallmark of cancer” do not solely refer to disseminating functions of tumor cells, but they also include cell growth, genome instability etc.. Please correct this sentence.
Main text
Line 66-70 – The authors need to acknowledge the recent discovery of the smaller extracellular vesicles exomeres.
Line 109 – Add exomeres to the figure.
Line 161-166 – The biogenesis of exomeres is not yet known, and it is not known if they derive from cell surface release. The authors should consider moving this paragraph to another section of the introduction.
Line 167-186 – This paragraph is very general and potentially belongs to more introductory sections of the manuscript. Some sentences are replicates or similar to the abstract and introduction. The authors might want to instead integrate some details of cellular uptake of extracellular vesicles involved in cell-cell communication.
Line 193-194 - The reference to the “hallmarks of cancer” is repeated too many times throughout the manuscript and is usually too general to be meaningful. Please refrain from replicating information.
Line 199 – The figure captions should me made bigger, as it is hard to read it in the current format.
Line 203-591 – The following sub-sections are in apparently random order. I suggest to order them according to cancer progression, starting from genome instability, to tumor cell proliferation and to metastasis.
Line 205-213 – Most of the listed strategies do not apply to exosome-mediated sustained growth and are not further developed in the text. Please shorten this section to only relevant information, and refer to other papers for a full list of other mechanisms.
Line 236-237 – This sentence and its reference belongs to the angiogenesis section, rather than 3.1 section.
Line 317-326 – The authors are trying here to show how exosomes promote metastasis in general, before delving into the mechanisms in the next paragraphs. However this section is very general, and omits to cite pivotal papers in the field, in particular on in vivo evidence. For example, previous work has shown that Rab27-KO in different cell lines dramatically reduce the ability of tumor cells to lead to metastasis in mouse models. These and other evidence should be acknowledged here. The authors need to expand and elaborate on further literature.
Line 327-358 – This section mostly focuses on the role of exosomes in inducing cell migration in vitro, which is hardly a real measurement of metastatic ability in vivo. Other aspects of the positive role of exosomes in inducing metastasis, such as vascular leakiness and ECM remodeling are minimally covered. The authors should refer to more relevant literature in the field, including in vivo and patient data, rather than focusing on in vitro cell migration studies.
Line 370-376 – This paragraph hardly belongs to this section, as it does not add anything to the topic of exosome and metastasis. Please remove or move to a different section.
Line 431-434 – The importance of hypoxia in EV-driven angiogenesis is questionable. Most of the papers cites in the next paragraph involve cultured tumor cells in normoxic condition, indicating that tumor cells can produce exosomes with pro-angiogenic cargo irrespective of hypoxia. Although the author should mention the evidence linking hypoxia and EV release, this might not be relevant for this section and should be removed or rephrased.
Line 459-474 – This section mainly focuses on the effect of EVs in propagating chemoresistance. As per “the hallmark of cancer”, other stimuli such as DNA damage in response to hyperproliferation can cause apoptosis. Tumor cells can circumvent apoptosis also by regulating apoptotic regulators and tumor suppressors (e.g. p53). The authors should make it clear if EVs have a more general role in the regulation of the apoptotic process, and discriminate this function to EV-induced chemoresistance.
Line 680 – Please expand “cols.” to “colleagues”.
Line 757-765 – There are some repetitions to the initial introductory paragraph of this section, please revise.
Line 782-788 – The authors omit to cite here a growing body of literature on the protein engineering of exosomes leading to increased/specific uptake. Please expand this section.
Line 802-803 – Please provide references for this claim. The authors should also elaborate on the fact that immune cells and fibroblasts are often described as main recipient of exosomes, and how this can be circumvented to deliver EVs directly to tumor cells as part of an imaging effort.
Line 877 – change to “designer”.
Line 889-891 – As commented before, exomeres should also be listed as a subtype of EVs.
Round 2
Reviewer 2 Report
The authors put sufficient effort to improve the manuscript and addressed most of my concerns in the revised version. I personally support the publication of the manuscript in its present form.
Reviewer 3 Report
With the added sections concerning the involvement of VE in the pathophysiology of cancer, this work currently seems to be more complete and it encompasses the maximum amount of data on this subject. The figures are well done and I consider that these figures are created by the authors. in the legends of these figures the meanings of the noted abbreviations are missing and must be added. Consequently, I consider that this work can be ready for publication.